# Mechanical Properties, Dry Shrinkage, and Water Penetration of Reusing Fine and Ultrafine Recycled Concrete Aggregate

**DOI:** 10.3390/ma15248947

**Published:** 2022-12-14

**Authors:** Tao Liu, Jianfeng Fan, Ziqiang Peng

**Affiliations:** 1School of Civil Engineering and Architectures, Wuhan University of Technology, Wuhan 430070, China; 2Department of Built Environment, Eindhoven University of Technology, 5600 MB Eindhoven, The Netherlands

**Keywords:** road construction, recycled concrete aggregate, mechanical preparties, shrinkage, water penetration

## Abstract

The effect of fine and ultrafine recycled concrete aggregate (RCA) on road construction still lacks investigation. This study investigates the properties of fine and ultrafine RCA, further, the influence of the different proportions of ultrafine RCA on the long-term performance of the designed matrix. The fine and ultrafine RCA are thoroughly characterized. Simultaneously, the mechanical properties, shrinkage properties, and water penetration of the designed matrix are evaluated. The results indicate that RCA shows low density, high porosity, and high water absorption. Furthermore, elevated ultrafine RCA contents result in higher compressive strength of up to 43.14% at 90 days and higher resistance of water penetration of up to 50% at 28 days due to the refined microstructure. However, higher drying shrinkage is observed with higher ultrafine RCA, which is associated with the high water absorption of the ultrafine RCA. The understanding of the utilization of ultrafine RCA in the construction of road base courses has been explored.

## 1. Introduction

Buildings, such as roads [1,2], bridges [3,4], and houses [5,6], require a large number of constructive resources. Utilizing construction and demolition waste (CDW) is crucial for the sustainable development and resource conservation of the building industry [7,8,9]. Currently, the RCA attracts increasing attention worldwide in construction [10,11]. However, the inferior properties of recycled concrete aggregate (RCA) limit its application in the engineering field [12].

The RCA is divided into coarse, fine, and ultrafine aggregates according to different grain sizes [13]. Previous studies showed that the high water absorption capacity of coarse RCA was due to the high porosity of adhered mortar in RCA [14,15,16]. According to the research, RCA with high water absorption had a significant impact on the development of mechanical and durability properties of concrete [17]. To date, the coarse RCA has been investigated intensively for application in road construction, replacing the natural aggregate [18,19]. However, the investigations about fine and ultrafine RCA focused on the chemical properties instead of the application scale [20]. Poon et al. [21] reported that the compositions of fine RCA were old cement paste and anhydrate cement grain containing C_2_S and C_3_S, which theoretically remained reactive. However, the complex properties of fine and ultrafine RCA, particularly ultrafine RCA, remain unknown problems applying in road base course construction [22]. The reasons are obvious: fine and ultrafine RCA exhibit significant heterogeneity, have a high capacity to absorb water and may contain toxins as a result of the recycling process. Previous studies [23,24] indicated that fine and ultrafine RCA with acceptable properties could be achieved by tuning the production process and performing pre-treatments. Then the improvement in properties could be obtained. However, those improvement procedures were created with coarse RCA, and it was unclear if they could be applied to fine and ultrafine RCA [24,25,26].

To discuss it further, the utilization of fine and ultrafine RCA in road base course construction should satisfy specific requirements, including mechanical properties [27], shrinkage [28], and resistance to water penetration [29]. The role of fine RCA in the CDW was investigated in terms of the mechanical properties of the mortars by Ruan et al. [30]. The increase in fineness of RCA contributes to improved mechanical properties due to its remaining reactivity. Besides, Wang et al. [31] reported the relationship between internal humidity and drying shrinkage of coarse RCA in concrete with different RCA content. Compared to natural aggregate, the RCA significantly influences the internal relative humidity and drying shrinkage of the concrete negatively. Moreover, Tran et al. [32] investigated the impact of coarse RCA on the water penetration of concrete. Because of the high coarse RCA porosity, concrete has a large water penetration and air permeability, which suggests that water and air can move through a porous network of the coarse RCA. In general, the effect of coarse RCA has been explored intensively. It is known that the particle size of RCA affects the properties of the matrix. Gao et al. [33] reported that the shear capacity of the concrete beam decreased as the replacement ratios of fine RCA and coarse RCA increased. However, the study about the impact of ultrafine RCA on the matrix is limited. Furthermore, fine and ultrafine RCA, especially ultrafine RCA, lacks investigations in terms of mechanical properties, shrinkage, and resistance to water penetration.

This study investigates the feasibility of reusing RCA in the road base course construction by different proportions of fine and ultrafine RCA in the matrix. The characterization was performed to evaluate the properties of raw materials. Then the mechanical and shrinkage properties of the matrix (fine RCA + ultrafine RCA + cement) were determined to understand the role of ultrafine RCA in the matrix. Subsequently, the water penetration of the samples was tested as well. The new insight into the role of ultrafine RCA in the application of road base course construction has been revealed.

## 2. Experimental Program

### 2.1. Starting Materials

The recycled concrete aggregate (RCA) (shown in Figure 1) originated from the Recycled Aggregate factory in Wuhan and was processed by a jaw crusher. The obtained aggregate was sieved to get fine aggregate (0.15~4.75 mm) and ultrafine aggregate (<0.15 mm). The basic testing of raw material was performed to evaluate the basic properties of RCA. The detailed testing methods are described in Section 2.3. Ordinary Portland cement (P·O 42.5 grade) used in this study meets the Chinese national standard GB/T 175-2007 [34].

### 2.2. Sample Preparations

All the testing mixtures were compressed (or compacted) in specific molds by a universal testing machine. The mix design for the construction of the road base course is shown in Table 1. The sizes of the samples are shown in Table 2.

The mass of the mixing water was calculated as follows:(1)$$m_{0}{=V\;\times \;\rho }_{\max}{\;\times \;(1+w}_{\mathrm{opt}})\;\times \;0.95$$
where $m_{0}$ is the sample mass; V is the sample volume; $\rho_{\max}$ is maximum dry density (MDD); $w_{\mathrm{opt}}$ is optimum moisture content (OMC); 0.95 is the extent of compaction.
(2)$$m_{1}=\frac{m_{0}}{{1+w}_{\mathrm{opt}}}$$
where $m_{1}$ is the drying mass of the sample.
(3)$$m_{w}{=m}_{1}{\;\times \;w}_{\mathrm{opt}}$$
where $m_{w}$ is the mass of mixing water.

### 2.3. Test Methods

The testing procedures are divided into two scales, including the properties of raw materials and the performance of the mixture (shown in Figure 2). The detailed experiment procedures are explained as follows.

#### 2.3.1. X-ray Diffractometry

X-ray diffractometry (XRD) was performed by using the Bruker D4 Phaser instrument (Bruker, Billerica, MA, USA)with Co-Kα radiation (40 kV, 30 mA). The pressed powdered specimens were measured with a step size of 0.1° and a counting time of 1 s/step, from 10° to 60° 2θ.

#### 2.3.2. X-ray Fluorescence

The chemical composition of ultrafine RCA was determined by X-ray fluorescence (XRF, PANalytical Epsilon 3). The XRF was performed by a fused bead sample. The fused beads were made using the borate fusion technique. The sample was blended with 10:1 borate flux and 100 μL of 4 M LiBr, which serves as a non-wetting agent. The ratio of Li_2_BO_7_ to LiBO_2_ in the borate flux was 67% to 33%. (obtained from Claisse, Malvern, UK). The liquid was homogenized before being heated to 1065 °C for 24 min in a borate fluxer oven (Classisse leNeo). The melt was then poured into a platinum mold to create a sample fused bead.

#### 2.3.3. Particle Size Distribution

The particle size distribution (PSD) of ultrafine RCA was determined by laser granulometry coupled with Master Sizer laser granulometry with an open measuring cell. The PSD curve of fine RCA was designed by adopting the Dinger-Funk equation as follows [35]:(4)$$U\left(D_{p}\right)=\frac{D_{p}^{n}-{\;D}_{\mathrm{ps}}^{n}}{D_{\mathrm{pl}}^{n}-{\;D}_{\mathrm{ps}}^{n}}$$
where: $U\left(D_{p}\right)$ is cumulative particle content under the sieve; $D_{p}^{n}$ is the particle size of the current particle; $D_{\mathrm{pl}}^{n}$ is the largest particle; $D_{\mathrm{ps}}^{n}$ is the particle size of the smallest particle; n is the distribution modulus.

The PSD of fine RCA was determined by sieving residues. Then the fineness modulus was calculated as follows:(5)$$M_{x}=\frac{A_{0.15}{+A}_{0.3}{+A}_{0.6}{+A}_{1.18}{+A}_{2.36}{+A}_{4.75}}{100}$$
where $M_{x}$ is the fineness modulus of fine RCA; $A_{0.15}$, $A_{0.3}$, $A_{0.6}$, $A_{1.18}$, $A_{2.36}$, and $A_{4.75}$ are the sieving residues by 0.15, 0.3, 0.6, 1.18, 2.36, and 4.75 mm sieves (unit: %), respectively.

#### 2.3.4. Aggregate Water Absorption Test

The aggregate water absorption test was performed by standard JTG E42—2005 [36]. The 1.5 kg fine aggregate was immersed in water for 24 h. Then the fine aggregate was filtered and placed in the open air until the dry surface state. The mass (m_1_) was recorded. Subsequently, the surface dry fine aggregate was placed in a drying oven at 105 °C for 24 h until constant weight. The mass (m_2_) was recorded as well. The test was repeated 3 times. The coefficient of water absorption (ω) was calculated as follows:(6)$$\omega =\frac{m_{1}{-m}_{2}}{m_{2}}\;\times \;100$$

#### 2.3.5. Bulk Density and Aggregate Porosity Test

The bulk density and aggregate porosity test was performed by Chinese standard GB/T 14684-2011 [37]. Bulk density ($\rho_{0}$): the 300 g (m_0_) of drying fine RCA was added to a 500 mL volumetric flask. Then fill the volumetric flask with water until the 500 mL scale. The total weight is m_1_. Then the volumetric flask was cleaned and filled only with water until the 500 mL scale. The weight of the volumetric flask and water is m_2_. The test was repeated 3 times. The bulk density was calculated as follows:(7)$$\rho_{0}=\frac{m_{0}}{m_{2\;}{+m}_{2}-m_{1}}{\;\times \;\rho }_{\mathrm{water}}$$

Loose bulk density($\rho_{1}$): the fine-drying RCA was added to a 300 mL volumetric flask until the fine RCA overflowed from the flask. Then using the spatula, scrape the flask. The volume of the flask is V, and the weight of the flask is G_1_. The total weight is G_2_.
(8)$$\rho_{1}=\frac{G_{2}-{\;G}_{1}}{V}$$

Loose porosity ($P_{1}$) was calculated as follows:(9)$$P_{1}=\left(1-\frac{\rho_{1}}{\rho_{0}}\right)\;\times \;100$$

The compacted bulk density ($\rho_{2}$): the drying fine RCA was added into a 300 mL volumetric flask with 25 times vibration to compact the aggregates. The fine RCA was added until the fine RCA overflowed from the flask. Then using the spatula scrape the flask. The volume of the flask is V, and the weight of the flask is G_1_. The total weight is G_2_.
(10)$$\rho_{2}=\frac{G_{2}-{\;G}_{1}}{V}$$

Compacted porosity ($P_{2}$) was calculated as follows:(11)$$P_{2}=\left(1-\frac{\rho_{2}}{\rho_{0}}\right)\;\times \;100$$

#### 2.3.6. Aggregate Crushing Test

For the aggregate crushing value (C) test as by [38], an aggregate sample passing through a 12.5 mm IS sieve and retained on a 10 mm IS sieve was selected and dried to a temperature of 105 °C to 110 °C then cooled to room temperature. To fill the cylindrical measure mold, about 6.5 kg of a sample of aggregate was sufficient.
(12)$$C=\frac{W_{3}}{W_{2\;}-{\;W}_{1}}\;\times \;100$$

W_1_ is the cylindrical measure’s empty weight. Then fill the measuring cylinder with the aggregate sample that passed through the 12.5 mm and retained on the 10 mm IS sieve in three equal layers, subjecting each layer to 25 tamping rod strokes. Consider W_2_ to be the aggregate weight as measured by the measuring cylinder. Fill a steel cylinder with a 15 cm diameter and 13 cm height with the aggregate sample, level the surface with a leveler, and then insert the plunger so that it sits horizontally on the surface. Use a compression device, and the load will be transferred to the aggregate in around 10 min. Release the pressure and take the steel cylinder out of the device. Take out the crushed aggregate sample and sieve on a 2.36 mm IS sieve; care is being taken to avoid loss of fines. Take the weight of the fraction passing through the 2.36 mm IS sieve as (W_3_).

#### 2.3.7. Proctor Compaction Test

To perform the proctor compaction test, 4 types of mixtures for the subbase are defined (U0, U4, U8, and U12). Every mixture with 2 kg weight was performed by proctor compaction test according to BS EN 13286 Part-2 [39]. Then the relationship between water content and dry density can be obtained.

#### 2.3.8. Unconfined Compressive Strength

On cured samples, the CRB-2 load ratio tester was used to evaluate the unconfined compressive strength at a preset strain rate of 1 mm/min. To assure the specimen’s smoothness before the test, the surface must be polished using sandpaper. Each group consisted of three specimens, and the typical strength value for this set of specimens was determined by taking the mean of three parallel samples.

#### 2.3.9. Splitting Tensile Strength

The specimen underwent the splitting tensile strength test at 28 curing days. The loading rate was set to 0.05 MPa/s, and the test was load-controlled. The three samples of the same mix design were examined at each temperature, and the recorded findings represent the average of the three readings.

#### 2.3.10. Resilient Modulus

The test was adopted by a universal testing machine according to standard JTG E51-2009 [40]. The deformation measuring device was placed on the universal testing machine. The sample was placed vertically on the bottom plate equipped with a dial indicator. The top plate of the testing machine was placed on the upper-end face of the sample; then, the universal testing machine was adjusted to make the loading device contact with the top surface of the sample. Preload was applied to ensure full contact with the samples. Then step-by-step loading and unloading measure the spring back deformation. The resilient modulus was calculated as follows:(13)$$E_{c}=\frac{\mathrm{ph}}{l}$$
where $E_{c}$ is the resilient modulus; p is the unit pressure; h is the height of the samples; l is the deformation of the samples.

#### 2.3.11. Water Loss and Drying Shrinkage

The testing samples were cast as 50 mm × 50 mm × 200 mm prisms. After demold, the samples were placed in a curing chamber at 20 °C and 95% humidity for 1 day. The initial weight and length of samples were recorded as $m_{p}$ (g) and l (mm), respectively. Then the samples were moved to a drying shrinkage chamber at 20 °C and 60% humidity. The weight ($m_{i}$) of every testing day was recorded as well as the drying shrinkage value ($\delta_{i}$). Subsequently, the water loss and drying shrinkage can be calculated as follows:(14)$$W_{i}=\frac{m_{i}-{\;m}_{i+1}}{m_{p}}$$
(15)$$\varepsilon_{i}=\frac{\delta_{i}}{l}$$

#### 2.3.12. Water Penetration

The water penetration test was performed by standard JTG-E30-2005 [41].The testing samples were cast in Φ150 mm × 150 mm as optimum water content and drying density determined by the proctor compaction test (shown in Section 3.1). Every group of the sample was cast 3 cylinders and then placed in a curing chamber at 20 °C and 95% humidity for specific curing days. After curing, the surface of the samples was wiped then the side surface of the samples was coated with paraffin wax until its drying. Subsequently, the samples were placed on a seepage meter. The loaded water pressure was constant at 0.8 MPa ± 0.05 MPa. The time was recorded as 0 when the water pressure was loaded. Then the water flow was recorded for 6 h. The coefficient of water penetration can be calculated as follow:(16)$$C_{w}=\frac{v_{2}-{\;v}_{1}}{t_{2}-{\;t}_{1}}\;\times \;60$$
where $C_{w}$ is the coefficient of water penetration (unit: mL/min); $v_{1}$ and $v_{2}$ are the water flow at the first and second time (unit: mL), respectively; $t_{1}$ and $t_{2}$ are the first and second times recording the water flow value (unit: s), respectively.

## 3. Results and Discussion

### 3.1. Characterization of RCA and Its Mixture

The characterization results of RCA and its mixture are shown in Figure 3 and Table 3 and Table 4. As can be seen from Figure 3a, high intensity of peak from quartz (SiO_2_, PDF: 01-086-1560) can be observed. It is from the old concrete aggregate, which is the main component of the RCA [42]. Meanwhile, the characteristic peak of calcite (CaCO_3_, PDF: 01-086-2334) can be observed at 34.3°. It is from the carbonation of Ca(OH)_2_ in the old concrete paste [43]. Simultaneously, a small amount of portlandite (Ca(OH)_2_, PDF: 01-081-2040) and albite ((Na,Ca)Al(Si,Al)_3_O_8_, PDF: 00-041-1480) are identified in the XRD patterns. The portlandite originates from the old concrete paste, which represents that the RCA is not completely carbonated [43]. The pozzolanic reaction between anhydrate cement paste and portlandite results in a further hydration process. This contributes to further refined microstructure and improved mechanical strength [43]. Besides, the albite is identified at 32.5°, which is attributed to the crystalline phases in the old concrete as well. It is in agreement with a previous study [44]. Table 3 shows the chemical compositions of ultrafine RCA. The primary contents of ultrafine RCA are Si and Ca up to 70 wt.%. The SiO_2_ content is the highest, which is due to the quartz from old aggregates.

Figure 3b,c exhibit the particle size distribution (PSD) of ultrafine and fine RCA. The d_50_ of ultrafine RCA is 29.94 μm. There is only 20% of super ultrafine RCA, lower than 10 μm. Therefore, the reactivity of the ultrafine RCA is relatively low. Because the smaller specific surface area of the ultrafine RCA represents lower reactivity, which depends on the PSD of ultrafine RCA [45]. Meanwhile, the fineness modulus of fine RCA is 3.45, which can be calculated by Equation (2). It corresponds to coarse sand [46]. The fineness modulus cannot accurately indicate the quality of their grading zones; it should be emphasized. The grading zones of sand with the same fineness modulus can vary greatly. As a result, the particle size and fineness modulus should be taken into account when preparing concrete. Furthermore, the PSD of ultrafine and fine RCA influences the workability of fresh concrete properties, the mechanical properties, shrinkage, and permeability of concrete, etc. [45]. It will be discussed in the following sections.

Table 4 presents the water absorption, bulk density, porosity, and crushing test of fine RCA. The water absorption of fine RCA is 9.69 wt. %, which is remarkably larger than the normal sand (1.52 wt. %), as reported in reference [47]. Old concrete sand in RCA has two typical forms: attached to the surface of aggregate and independent existence [48]. In this study, the old concrete sand exhibits an independent existence in RCA. It leads to large porosity of RCA. At the same time, there are a huge number of micro-cracks generated during the RCA crushing process [49]. Therefore, both reasons increase the water absorption of fine RCA. Besides, the bulk density, loose bulk density, and compacted bulk density of fine RCA are 2430, 1245, and 1380 kg/m^3^, respectively. The low densities result from the irregular shape of fine RCA, leading to large loose (48.7%) and compacted (43.2%) porosities. In addition, the crashing value of fine RCA is 22%; it is due to a large amount of loose old cement paste in fine RCA. Consequently, micro-cracks and other internal damage cause a high crashing value [49].

Figure 3d shows the dry densities and corresponding water contents curves of the mixtures (shown in Table 1). The optimum moisture content (OMC) and maximum dry density (MDD) of the samples can be derived by the curves. As can be seen, the OMCs of U0, U4, U8, and U12 are 8.7, 9.8, 10.4, and 11.5%, respectively. The OMC increases with higher ultrafine RCA percentages. This is attributed to the high water absorption of ultrafine RCA, and the water remains between the particles adhered to the particles. Simultaneously, the MDD of U0, U4, U8, and U12 are 1.875, 1.909, 1.918, and 1.928 g/cm^−3^, respectively. The MDD also increases with the higher ultrafine RCA percentages. Because the filling effect of ultrafine RCA leads to denser particle packing of the mix, it should be noted that RCA mixes should contain some free water to achieve the desired efficient compaction in OMC, which will be consumed by the cement hydration process [50].

### 3.2. Mechanical Properties

Figure 4 illustrates the mechanical properties of samples including unconfined compressive strength, splitting tensile strength, resilient modulus, and unconfined compressive strength vs. resilient modulus. Firstly, the average unconfined compressive strength of three samples with the error bar in 7, 28, and 90 days is shown in Figure 4a. It is obvious that the strength increases with the higher content of ultrafine RCA in the matrix. Furthermore, U12 shows the highest strength during the curing ages. Compared with U0, the compressive strength of U12 increased by 65.63, 51.22, and 43.14% at 7, 28, and 90 days, respectively. It is associated with the remaining activity of anhydrate cement in RCA [51,52]. The finer RCA provides higher activity under the curing condition forming more C–S–H gels. Simultaneously, the ultrafine RCA absorbs more water during the mixing state. It agrees with OMC testing results. Then more water can be absorbed by more ultrafine RCA contents promoting further hydration during the curing process of the samples [53]. Thus the remaining water in the samples promotes strength in the late hydration period. Therefore, the higher ultrafine RCA in the matrix shows better performance in long-term compressive strength.

Similarly, the splitting tensile strength also increases with more ultrafine RCA contents in the sample along curing ages (shown in Figure 4b). All the samples reach 0.3~0.4 MPa, 0.6~0.8 MPa, and 0.8~1 MPa on days 7, 28, and 90, respectively. Meanwhile, the more ultrafine RCA contents contribute to higher splitting tensile strength. The splitting tensile strengths of U0, U4, U8, and U12 are 0.67, 0.71, 0.79, and 0.85 MPa at day 28, respectively. Compared to U0, the strength of U4, U8, and U12 increased by 5.97, 17.91, and 26.87%, respectively. Besides, the splitting tensile strengths of U0, U4, U8, and U12 are 0.83, 0.95, 1.05, and 1.06 MPa at day 90, respectively. The strength increases of U4, U8, and U12 are 14.46, 26.51, and 27.71% compared with U0, respectively. It can be observed that a longer curing time brings more strength development because the ultrafine RCA is beneficial to strength development due to its relatively higher reactivity towards gel formation [54]. The reactivity of ultrafine RCA shows the influence on reaction in the longer curing ages. This is in line with the results of unconfined compressive strength.

Figure 4c,d illustrates the resilient modulus at day 90 and the fitting of unconfined compressive strength vs. resilient modulus at day 90. The resilient modulus of U0, U4, U8, and U12 are 872, 1135, 1277, and 1429 MPa at day 90, respectively. With the elevated content of ultrafine RCA, the increases of resilient modulus in U4, U8, and U12 are 30.16, 46.44, and 63.88%, respectively. As far as the resilient modulus vs. unconfined compressive strength is concerned in Figure 4d, it can be observed that the resilient modulus is linear with unconfined compressive strength. R^2^ of its fitting equation is 0.9519, which indicates that there is a very strong linear relationship between resilient modulus and unconfined compressive strength. It is in agreement with the reference [55].

### 3.3. Shrinkage Properties

Figure 5 illustrates the shrinkage properties of the samples along the curing ages, including water loss, drying shrinkage, coefficient of shrinkage, and the relationship between water loss and drying shrinkage. As can be seen in Figure 5a, the water loss of the samples increases dramatically between the first two days. However, the drying shrinkage shows limited changes in this period (shown in Figure 5b). Because a large amount of free water in the pores evaporates at the beginning stages [56], as a consequence, limited drying shrinkage is observed. Further water loss can be observed during the testing ages. It significantly affects the drying shrinkage at the late stage. Notably, water loss decreases with the higher ultrafine RCA content (shown in Figure 5a). It is due to the large water absorption of ultrafine RCA powder. However, the higher ultrafine RCA contents lead to higher drying shrinkage at the late curing ages (shown in Figure 5b). It is obvious that the water loss and drying shrinkage increase with the testing days. Simultaneously, the water loss and drying shrinkage change remarkably at the early curing ages and stabilize at late curing ages. The water loss reaches 70~80% of the total water loss within 7 days, while the drying shrinkage reaches 80~90% of total drying shrinkage within 14 days. This is associated with the available free water in the pore solution.

Figure 5c exhibits the increasing coefficients of shrinkage along the curing ages. The 28-day coefficient of U0 shows the minimum value (59.94 × 10^−6^) among the samples, while the coefficient of U8 shows the maximum value (90.77 × 10^−6^). The U12 shows a lower coefficient of shrinkage than that of U4 and U8. It may be due to the higher water absorption of ultrafine RCA releasing the free water in the late curing ages as the internal curing effect [31]. Then the interfacial transition zone is improved by the internal curing effect, resulting in limited shrinkage and a lower coefficient of shrinkage.

Figure 5d illustrates the relationship between water loss and drying shrinkage by regression analysis along the curing ages. As can be seen, the R^2^ of all fitting equations is higher than 0.9. It represents that drying shrinkage is highly correlated with water loss. The drying shrinkage of the higher ultrafine RCA leads to more correlation with water loss. Because the ultrafine RCA can absorb and maintain more free water during the mixing process.

### 3.4. Resistance of Water Penetration

Figure 6 shows that the water penetration coefficient of samples decreases with the ultrafine RCA contents at 7 and 28 days. The coefficients of water penetration in U0, U4, U8, and U12 are 38.9, 34.1, 32.7, and 26.2 mL/(min × 10^−2^) at day 7, respectively. Compared to U0, the water penetration coefficients of U4, U8, and U12 decrease 12.34, 15.94, 32.65%, respectively. In addition, the coefficients of water penetration in U0, U4, U8, and U12 are 33.6, 28.5, 24.1, and 16.8 mL/(min × 10^−2^) at day 28, respectively. Compared to U0, the water penetration coefficients of U4, U8, and U12 decrease 15.18, 28.27, 50%, respectively. It is associated with the better particle packing of the matrix by the higher incorporation of ultrafine RCA. Meanwhile, the anhydrate cement in ultrafine RCA promotes the formation of gels, leading to a denser microstructure of the matrix [52,57]. It results in a lower water penetration coefficient as well. Besides, the water penetration coefficient of samples decreases with curing ages as well, which is also due to the hydration products formed from anhydrate cement in RCA. It can be seen from the results that higher ultrafine RCA contents reduce coefficients of water penetration in the late curing ages. Because the higher ultrafine RCA significantly affects the late hydration process of RCA at the late curing period in the matrix.

## 4. Conclusions

This study investigates the effect of ultrafine and fine recycled concrete aggregate (RCA) on the construction of road base courses on the lab scale. It shows that the higher proportion of ultrafine RCA promotes the mechanical properties and long-term durability of the matrix due to its relatively high reactivity. The following detailed conclusions can be drawn:The particle size distribution indicates the fine RCA belonging to coarse sand. Meanwhile, RCA shows low density, high porosity, and high water absorption.More ultrafine RCA leads to an increase in optimum moisture content and maximum dry density of the mixture, which is due to its higher water absorption than the fine RCA.Elevated ultrafine RCA contents result in higher compressive strength (up to 43.14%), splitting tensile strength (up to 27.71%), and resilient modulus (up to 63.88%) at 90 days. It is associated with the higher reactivity of ultrafine RCA than fine RCA.More ultrafine RCA increases the drying shrinkage during its high water absorption and more water loss during the curing period. Simultaneously, the drying shrinkage is highly correlated with water loss (R^2^ > 0.9 in all cases).Higher ultrafine RCA contents show higher resistance to water penetration because ultrafine RCA promotes gel formation, which improves the microstructure of the matrix.

Recommendation of the research: the optimum ratio of fine and ultrafine RCA in road base course construction needs more investigations, including wet-dry cycling, abrasion resistance, thermal conductivity, etc. Furthermore, a practical case study can be performed by combining lab-scale experiments.

## Figures and Tables

**Figure 1 materials-15-08947-f001:**
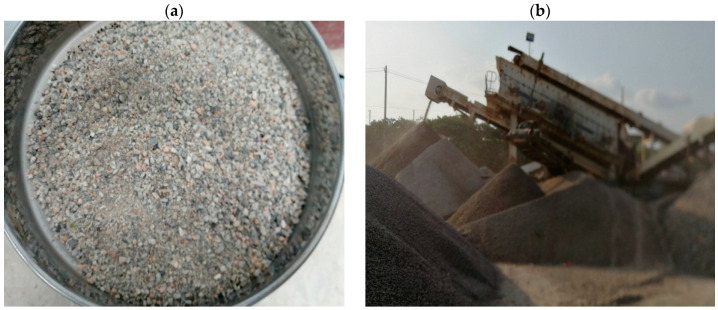
(**a**) Raw recycled concrete aggregate; (**b**) aggregate production.

**Figure 2 materials-15-08947-f002:**
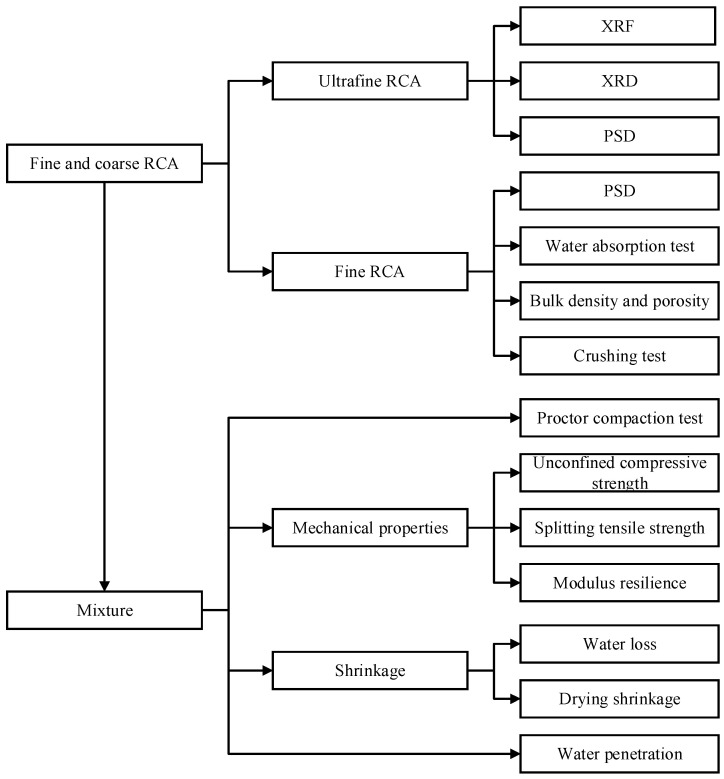
Testing procedures.

**Figure 3 materials-15-08947-f003:**
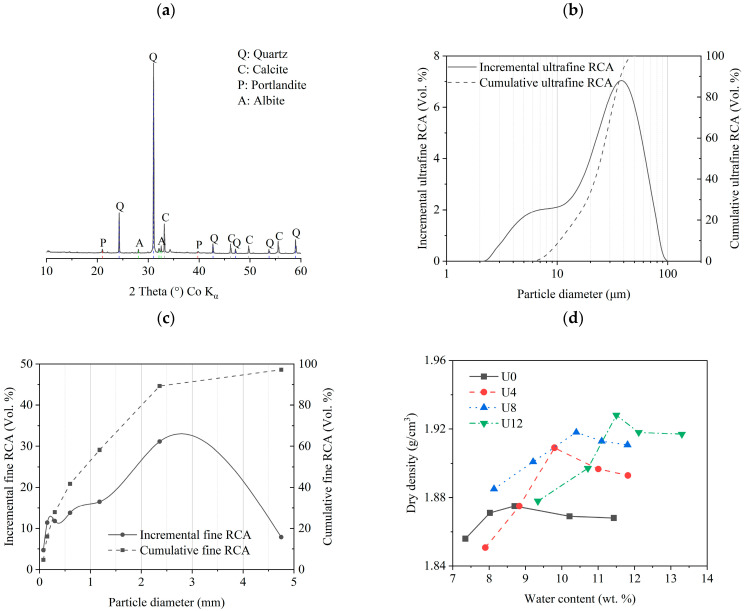
(**a**) XRD pattern of ultrafine RCA; (**b**) PSD of ultrafine RCA; (**c**) PSD of fine RCA; (**d**) Optimum water content and maximum drying density.

**Figure 4 materials-15-08947-f004:**
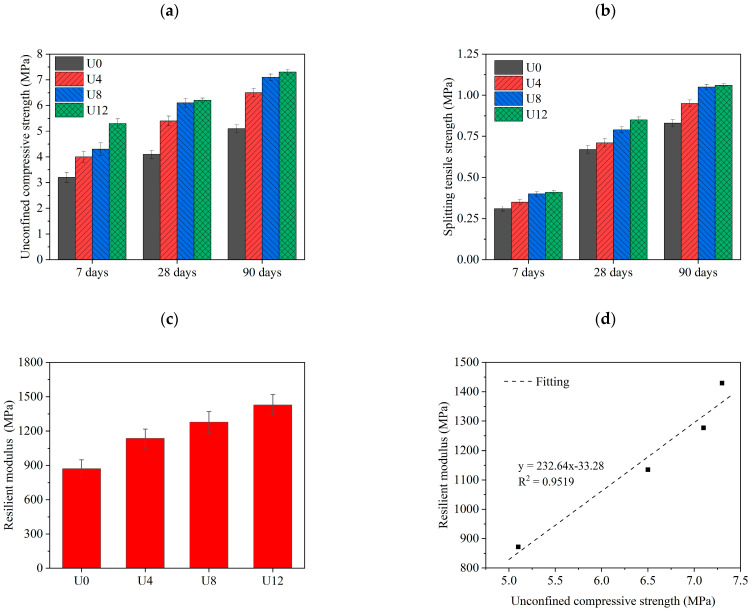
Mechanical properties (**a**) Unconfined compressive strength; (**b**) Splitting tensile strength; (**c**) Resilient modulus at 90 days; (**d**) Fitting of unconfined compressive strength vs. resilient modulus at 90 days.

**Figure 5 materials-15-08947-f005:**
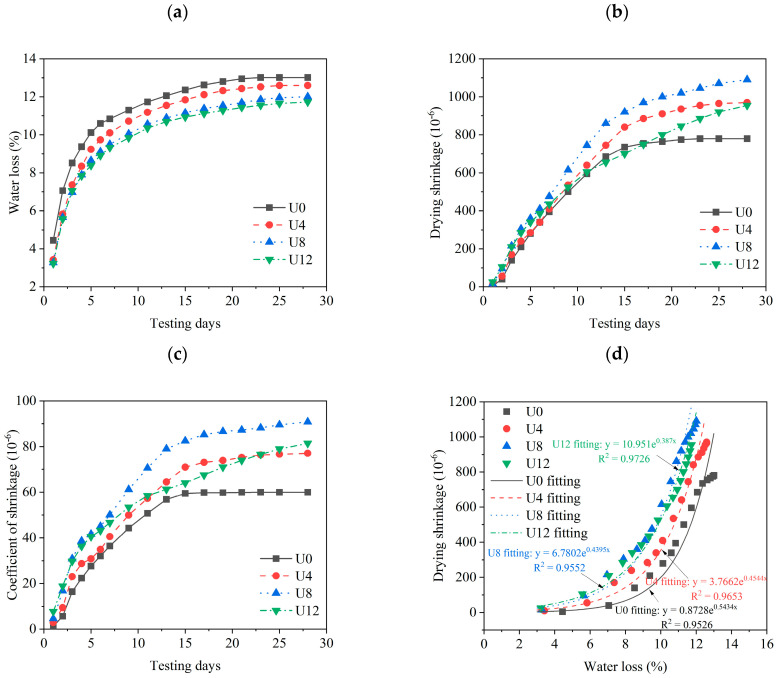
(**a**) Water loss; (**b**) Drying shrinkage; (**c**) Coefficient of shrinkage; (**d**) Relationship between water loss and drying shrinkage. U0 fitting: y = 0.8728e^0.5434x^, R^2^ = 0.9526; U4 fitting: y = 3.7662e^0.4544x^, R^2^ = 0.9653; U8 fitting: y = 6.7802e^0.4395x^, R^2^ = 0.9552; U12 fitting: y = 10.951e^0.387x^, R^2^ = 0.9726.

**Figure 6 materials-15-08947-f006:**
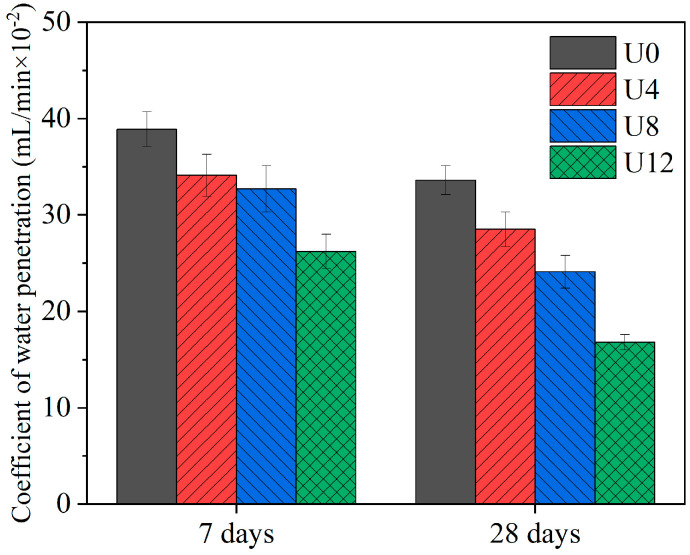
Coefficient of water penetration at 7 and 28 days.

**Table 1 materials-15-08947-t001:** Mix design of samples (wt. %).

Sample ID	Cement	Ultrafine Aggregate	Fine Aggregate
U0	5	0	100
U4		4	96
U8		8	92
U12		12	88

**Table 2 materials-15-08947-t002:** Size of samples.

Testing	Sample Size
Unconfined compressive strength	Φ50 mm × 50 mm
Splitting tensile strength	
Resilient modulus	Φ100 mm × 100 mm
Shrinkage testing	50 mm × 50 mm × 200 mm
Water penetration	Φ150 mm × 150 mm

**Table 3 materials-15-08947-t003:** Chemical composition (wt. %) of ultrafine RCA.

	SiO_2_	CaO	Al_2_O_3_	Fe_2_O_3_	Na_2_O	K_2_O	SO_3_	MgO	Cl
Ultrafine RCA	49.75	22.31	10.66	4.12	2.10	1.97	1.88	1.62	0.07

**Table 4 materials-15-08947-t004:** Water absorption, bulk density, porosity, and crushing test of fine RCA.

	Coefficient of Water Absorption	Bulk Density	Loose Bulk Density	Compacted Bulk Density	Loose Porosity	Compacted Porosity	Crushing Value
Unit	wt. %	kg/m^3^	kg/m^3^	kg/m^3^	%	%	%
Fine RCA	9.69	2430	1245	1380	48.7	43.2	22

## Data Availability

The data presented in this study are available from the first author upon reasonable request.

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
