# Peer review of "Mechanical Properties, Dry Shrinkage, and Water Penetration of Reusing Fine and Ultrafine Recycled Concrete Aggregate"

_materials, 2022, doi:10.3390/ma15248947_

Round 1
Reviewer 1 Report
Line 75: are instead of were?
line 78: were instead of was?
It could be interesting to mention where the equations come from, according to a paper, book, or standard.
Line 214 - 215: coherence in the sentence. Figure 2 and Table 3, show characterization results, but Table 2?
Line 224: words missing at the beginning of the sentence. So, it has sense.
Line 214 - 225: Suggestion.... After the crystallinity and composition shown in Figure 2a and Table 2, what kind of structures or compositions could be expected that promote and ensure the mechanical properties obtained? This could be the best contribution regarding the microstructure formation assumption. This approach is somehow confirmed in sections 3.2, 3.3.
Line 303: linear instead of liner?
Legends or labels in figure 4d, are blurry, perhaps, writing the y´s and R´s in the figure caption.
Author Response
Comments and reply: All the replies in the following response and revised paper are in red font.
Line 75: are instead of were?
Reply: We thank the reviewer for the careful observation. It is revised. Line 78.
line 78: were instead of was?
Reply: We thank the reviewer for the careful observation. It is revised. Line 83.
It could be interesting to mention where the equations come from, according to a paper, book, or standard.
Reply: We thank the reviewer for the constructive comment. All the relevant standards have been cited in the manuscript.
Line 214 - 215: coherence in the sentence. Figure 2 and Table 3, show characterization results, but Table 2?
Reply: We are sorry for the misunderstanding. It should be Figure 2, Table 3, and Table 4 show characterization results. It has been revised. Line 223 and 224.
Line 224: words missing at the beginning of the sentence. So, it has sense.
Reply: We are sorry for the misunderstanding. It should be table 3. It is revised. Line 235.
Line 214 - 225: Suggestion.... After the crystallinity and composition shown in Figure 2a and Table 2, what kind of structures or compositions could be expected that promote and ensure the mechanical properties obtained? This could be the best contribution regarding the microstructure formation assumption. This approach is somehow confirmed in sections 3.2, 3.3.
Reply: Thanks for the suggestion. The corresponding description has been added: The pozzolanic reaction between anhydrate cement paste and portlandite results in a further hydration process. This contributes to further refined microstructure and im-proved mechanical strength. Line 231-223.
Line 303: linear instead of liner?
Reply: It is revised. Line 314.
Legends or labels in figure 4d, are blurry, perhaps, writing the y´s and R´s in the figure caption.
Reply: We thanks for the careful observation from the reviewer. All the fitting equations are added in the figure 4d caption.
Reviewer 2 Report
This manuscript evaluates the “Mechanical properties, dry shrinkage, and water penetration of reusing fine and ultrafine recycled concrete aggregate”. The manuscript is described and contextualized with the help of previous and present theoretical background. However it needs elaboration. All the references cited are relevant to this area of research. The methods/analytical study are clearly stated. The result and discussion section are clearly presented. The manuscript needs major revision and require the following modifications before the acceptance.
1. Abstract is more general. It should be specific, indicating your results. For example, the percentage increase in compressive strength should be mentioned. Revise the abstract so that it reflect your results in specific.
2. The sentence ‘Utilizing construction and demolition waste (CDW) is crucial for the sustainable development and resource conservation of the building industry’ need to be cited with some more works. Some works are below.
https://doi.org/10.1002/suco.201800355
https://doi.org/10.12989/sem.2022.83.3.387
3. Introduction need elaboration.
4. Show the pictures of Materials used in this research and also experimental photos.
5. Test methods are too elaborated. Cut the procedure of all the methods. Just say the guidelines/relevant standads/code.. It is enough.
6. Table 3. Correct the font size of the content
7. Present the recommendation of your research at the end of the conclusions.
Author Response
Comments and reply: All the replies in the following response and revised paper are in red font.
- Abstract is more general. It should be specific, indicating your results. For example, the percentage increase in compressive strength should be mentioned. Revise the abstract so that it reflect your results in specific.
Reply: We thank the reviewer for the constructive comment. The specific increase of strength and higher resistance of water penetration have been added in the abstract. Line 14-16.
- The sentence ‘Utilizing construction and demolition waste (CDW) is crucial for the sustainable development and resource conservation of the building industry’ need to be cited with some more works. Some works are below.
https://doi.org/10.1002/suco.201800355
https://doi.org/10.12989/sem.2022.83.3.387
Reply: We thank the reviewer for the suggested works. The papers have been cited in the introduction to support the description. Line 25-26.
- Introduction need elaboration.
Reply: : We thank the reviewer for the constructive comment.. The gap about fine and ultrafine RCA has been added in the manuscript. Line 59-64.
- Show the pictures of Materials used in this research and also experimental photos.
Reply: Thank the reviewer for the comment. The pictures of raw materials and aggregate production have been added. However, the quality of the experimental photos are in low quality, so we didn’t add in the manuscript. Figure 1.
- Test methods are too elaborated. Cut the procedure of all the methods. Just say the guidelines/relevant standads/code.. It is enough.
Reply: We thank the reviewer for the comment Some methods followed by Chinese standards, so we described it in detail. The relevant standards have been cited as well.
- Table 3. Correct the font size of the content
Reply: We thank the reviewer for the careful observation. The font size has been adjusted. Table 3.
- Present the recommendation of your research at the end of the conclusions.
Reply: We thank the reviewer for the constructive comment. The detailed description has been added: Recommendation of the research: the optimum ratio of fine and ultrafine RCA in road base course construction needs more investigations including wet-dry cycling, abrasion resistance, thermal conductivity, etc. Furthermore, a practical case study can be performed by combining lab-scale experiments. Line 387-390.
Reviewer 3 Report
The topic of the paper is interesting but the novelty of the paper is missing. The following is required to be considered by the authors before the paper being accepted.
· The gap in the literature has not been stated and the novelty of the current research is missing. Therefore, these must be addressed in the revised version.
Author Response
Comments and reply: All the replies in the following response and revised paper are in red font.
The gap in the literature has not been stated and the novelty of the current research is missing. Therefore, these must be addressed in the revised version.
Reply: We thank the reviewer for the constructive comment.. The gap about fine and ultrafine RCA has been added in the manuscript. Line 59-64.
Reviewer 4 Report
1. Corrections needed in the manuscript, the correct words to be included are provided.

Author Response
We thank the reviewer for the constructive comments. The manuscript has been revised according to the comments. All the revisions are in red font
Round 2
Reviewer 2 Report
The author addressed all the comments well. It can accepted in the present form.